# Visual Adversarial Imitation Learning using Variational Models

**Rafael Rafailov**[1]    **Tianhe Yu**[1]    **Aravind Rajeswaran**[2,3]    **Chelsea Finn**[1]
{rafailov, tianheyu, cbfinn}@stanford.edu, aravraj@fb.com
[1] Stanford University, [2] University of Washington, [3] Facebook AI Research

## Abstract

Reward function specification, which requires considerable human effort and iteration, remains a major impediment for learning behaviors through deep reinforcement learning. In contrast, providing visual demonstrations of desired behaviors often presents an easier and more natural way to teach agents. We consider a setting where an agent is provided a fixed dataset of visual demonstrations illustrating how to perform a task, and must learn to solve the task using the provided demonstrations and unsupervised environment interactions. This setting presents a number of challenges including representation learning for visual observations, sample complexity due to high dimensional spaces, and learning instability due to the lack of a fixed reward or learning signal. Towards addressing these challenges, we develop a variational model-based adversarial imitation learning (V-MAIL) algorithm. The model-based approach provides a strong signal for representation learning, enables sample efficiency, and improves the stability of adversarial training by enabling on-policy learning. Through experiments involving several vision-based locomotion and manipulation tasks, we find that V-MAIL learns successful visuomotor policies in a sample-efficient manner, has better stability compared to prior work, and also achieves higher asymptotic performance. We further find that by transferring the learned models, V-MAIL can learn new tasks from visual demonstrations without any additional environment interactions. All results including videos can be found online at https://sites.google.com/view/variational-mail.

## 1   Introduction

The ability of reinforcement learning (RL) agents to autonomously learn by interacting with the environment presents a promising approach for learning diverse skills. However, reward specification has remained a major challenge in the deployment of RL in practical settings [1, 2, 3]. The ability to imitate humans or other expert trajectories allows us to avoid the reward specification problem, while also circumventing challenges related to task-specific exploration in RL. Visual demonstrations can also be a more natural way to teach robots various tasks and skills in real-world applications. However, this setting is also fraught with a number of technical challenges including representation learning for visual observations, sample complexity due to the high dimensional observation spaces, and learning instability [4, 5, 6] due to lack of a stationary learning signal. We aim to overcome these challenges and to develop an algorithm that can learn from limited demonstration data and scale to high-dimensional observation and action spaces often encountered in robotics applications.

Behaviour cloning (BC) is a classic algorithm to imitate expert demonstrations [7], which uses supervised learning to greedily match the expert behaviour at demonstrated expert states. Due to environment stochasticity, covariate shift, and policy approximation error, the agent may drift away from the expert state distribution and ultimately fail to mimic the demonstrator [8]. While a wide initial state distribution [9] or the ability to interactively query the expert policy [8] can circumvent

35th Conference on Neural Information Processing Systems (NeurIPS 2021).

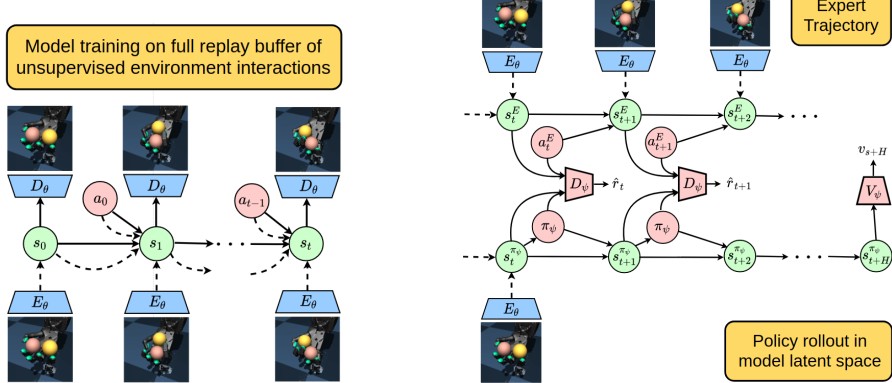

Figure 1: **Left**: the variational dynamics model, which enables joint representation learning from visual inputs and a latent space dynamics model, and the discriminator which is trained to distinguish latent states of expert demonstrations from that of policy rollouts. Dashed lines represent inference and solid lines represent the generative model. **Right**: the policy training, which uses the discriminator as the reward function, so that the policy induces a latent state visitation distribution that is indistinguishable from that of the expert. The learned policy network is composed with the image encoder from the variational model to recover a visuomotor policy.

these difficulties, such conditions require additional supervision and are difficult to meet in practical applications. An alternate line of work based on inverse RL [10, 11] and adversarial imitation learning [12, 13] aims to not only match actions at demonstrated states, but also the long term state visitation distribution of the expert [14]. Adversarial imitation learning approaches explicitly train a GAN-based classifier [15] to distinguish the visitation distribution of the agent from the expert, and use it as a reward signal for training the agent with RL. While these methods have achieved substantial improvement over behaviour cloning without additional expert supervision, they are difficult to deploy in realistic scenarios for multiple reasons: (1) the objective requires on-policy data collection leading to high sample complexity; (2) the reward function changes as the RL agent learns; and (3) high-dimensional observation spaces require representation learning and exacerbate the optimization challenges.

Our main contribution in this work is the development of a new algorithm, variational model-based adversarial imitation learning (V-MAIL), which aims to overcome each of the aforementioned challenges within a single framework. As illustrated in Figure 1, V-MAIL trains a variational latent-space dynamics model and a discriminator that provides a learning reward signal by distinguishing latent rollouts of the agent from the expert. The key insight of our approach is that variational models can address these challenges simultaneously by (a) making it possible to collect on-policy roll-outs inside the model without environment interaction, leading to an efficient and stable optimization process and (b) providing a rich auxiliary objective for efficiently learning compact state representations and which regularizes the discriminator. Furthermore, the variational model also allows V-MAIL to perform zero-shot transfer to new imitation learning tasks. By generating on-policy rollouts within the model, and training the discriminator using these rollouts along with demonstrations of a new task, V-MAIL can learn policies for new tasks without any additional environment interactions.

Through experiments on vision-based locomotion and manipulation tasks, we find that V-MAIL can successfully learn visuomotor control policies from limited demonstrations. In particular, V-MAIL exhibits stable and near-monotonic learning, is highly sample efficient, and asymptotically matches the expert level performance on most tasks. In contrast, prior algorithms exhibit unstable learning and poor asymptotic performance, often achieving less that 20% of expert performance on these vision-based tasks. We further show the ability to transfer the model to novel tasks, acquiring qualitatively new behaviors using only a few demonstrations and no additional environment interactions.

## 2   Preliminaries

We consider the problem setting of learning in partially observed Markov decision processes (POMDPs), which can be described with the tuple: $\mathcal{M} = (\mathcal{S}, \mathcal{A}, \mathcal{X}, \mathcal{R}, \mathcal{T}, \mathcal{U}, \gamma)$, where $s \in \mathcal{S}$

is the state space, $\boldsymbol{a} \in \mathcal{A}$ is the action space, $\boldsymbol{x} \in \mathcal{X}$ is the observation space and $r = \mathcal{R}(\boldsymbol{s}, \boldsymbol{a})$ is a reward function. The state evolution is Markovian and governed by the dynamics as $\boldsymbol{s}' \sim \mathcal{T}(\cdot|\boldsymbol{s}, \boldsymbol{a})$. Finally, the observations are generated through the observation model $\boldsymbol{x} \sim \mathcal{U}(\cdot|\boldsymbol{s})$. The widely studied Markov decision process (MDP) is a special case of this 7-tuple where the underlying state is directly observed in the observation model.

In this work, we study imitation learning in unknown POMDPs. Thus, we do not have access to the underlying dynamics, the true state representation of the POMDP, or the reward function. In place of the rewards, the agent is provided with a fixed set of expert demonstrations collected by executing an expert policy $\pi^E$, which we assume is optimal under the unknown reward function. The agent can interact with the environment and must learn a policy $\pi(\boldsymbol{a}_t|\boldsymbol{x}_{\leq t})$ that mimics the expert.

## 2.1 Imitation learning as divergence minimization

In line with prior work, we interpret imitation learning as a divergence minimization problem [12, 14, 16]. For simplicity of exposition, we consider the MDP case in this section, and discuss POMDP extensions in Section 3.2. Let $\rho^\pi_{\mathcal{M}}(\boldsymbol{s}, \boldsymbol{a}) = (1 - \gamma) \sum_{t=0}^{\infty} \gamma^t P(\boldsymbol{s}_t = \boldsymbol{s}, \boldsymbol{a}_t = \boldsymbol{a})$ be the discounted state-action visitation distribution of a policy $\pi$ in MDP $\mathcal{M}$. Then, a divergence minimization objective for imitation learning corresponds to

$$\min_{\pi} \; \mathbb{D}(\rho^\pi_{\mathcal{M}}, \rho^E_{\mathcal{M}}), \tag{1}$$

where $\rho^E_{\mathcal{M}}$ is the discounted visitation distribution of the expert policy $\pi^E$, and $\mathbb{D}$ is a divergence measure between probability distributions such as KL-divergence, Jensen-Shannon divergence, or a generic $f-$divergence. To see why this is a reasonable objective, let $J(\pi, \mathcal{M})$ denote the expected value of a policy $\pi$ in $\mathcal{M}$. Inverse RL [17, 12, 13] interprets the expert as the optimal policy under some unknown reward function. With respect to this unknown reward function, the sub-optimality of any policy $\pi$ can be bounded as:

$$\left| J(\pi^E, \mathcal{M}) - J(\pi, \mathcal{M}) \right| \leq \frac{R_{\max}}{1 - \gamma} \, \mathbb{D}_{TV}(\rho^\pi_{\mathcal{M}}, \rho^E_{\mathcal{M}}),$$

since the policy performance is $(1 - \gamma) \cdot J(\pi, \mathcal{M}) = \mathbb{E}_{(\boldsymbol{s}, \boldsymbol{a}) \sim \rho^\pi_{\mathcal{M}}} \left[ r(\boldsymbol{s}, \boldsymbol{a}) \right]$. We use $\mathbb{D}_{TV}$ to denote total variation distance. Since various divergence measures are related to the total variation distance, optimizing the divergence between visitation distributions in state space amounts to optimizing a bound on the policy sub-optimality.

## 2.2 Generative Adversarial Imitation Learning (GAIL)

With the divergence minimization viewpoint, any standard generative modeling technique including density estimation, VAEs, GANs etc. can in principle be used to minimize Eq. 1. However, in practice, use of certain generative modeling techniques can be difficult. A standard density estimation technique would involve directly parameterizing $\rho^\pi_{\mathcal{M}}$, say through auto-regressive flows, and learning the density model. However, a policy that induces the learned visitation distribution in $\mathcal{M}$ is not guaranteed to exist and may prove hard to recover. Similar challenges prevent the direct application of a VAE based generative model as well. In contrast, GANs allow for a policy based parameterization, since it only requires the ability to sample from the generative model and does not require the likelihood. This approach was followed in GAIL, leading to the optimization

$$\max_{\pi} \min_{D_\psi} \mathbb{E}_{(\boldsymbol{s}, \boldsymbol{a}) \sim \rho^E_{\mathcal{M}}} \left[ -\log D_\psi(\boldsymbol{s}, \boldsymbol{a}) \right] \; + \; \mathbb{E}_{(\boldsymbol{s}, \boldsymbol{a}) \sim \rho^\pi_{\mathcal{M}}} \left[ -\log \left( 1 - D_\psi(\boldsymbol{s}, \boldsymbol{a}) \right) \right], \tag{2}$$

where $D_\psi$ is a discriminative classifier used to distinguish between samples from the expert distribution and the policy generated distribution. Results from Goodfellow et al. [15] and Ho and Ermon [12] suggest that the learning objective in Eq. 2 corresponds to the divergence minimization objective in Eq. 1 with Jensen-Shannon divergence. In order to estimate the second expectation in Eq. 2 we require on-policy samples from $\pi$, which is often data-inefficient and difficult to scale to high-dimensional image observations. Some off-policy algorithms [18, 19] replace the expectation under the policy distribution with expectation under the current replay buffer distribution, which allows for off-policy training, but can no longer guarantee that the induced visitation distribution of the learned policy will match that of the expert.

# 3 Variational Model-Based Adversarial Imitation Learning

Imitation learning methods based on expert distribution matching have unique challenges. Improving the generative distribution of trajectories (through policy optimization, as we do not have control over the environment dynamics) requires samples from $\rho_{\mathcal{M}}^{\pi}$, which requires rolling out $\pi$ in the environment. Furthermore, the optimization landscape of a saddle point problem (see Eq. 2) can require many iterations of learning, each requiring fresh on-policy rollouts. This is different from typical generative modeling applications [15, 20] where sampling from the generator is cheap. To overcome these challenges, we present a model-based imitation learning algorithm. Model-based algorithms can utilize a large number of *synthetic on-policy* rollouts using the learned dynamics model, with periodic model correction. In addition, learning the dynamics model serves as a rich auxiliary task for state representation learning, making policy learning easier and more sample efficient. For conceptual clarity and ease of exposition, we first present our conceptual algorithm in the MDP setting in Section 3.1, and then extend this algorithm to the POMDP case in Section 3.2. Finally, we present a practical version of our algorithm in Sections 3.3 and 3.4.

## 3.1 Model-Based Adversarial Imitation Learning

Model-based algorithms for RL and IL involve learning an approximate dynamics model $\widehat{\mathcal{T}}$ using environment interactions. The learned dynamics model can be used to construct an approximate MDP $\widehat{\mathcal{M}}$. In our context of imitation learning, learning a dynamics model allows us to generate samples from $\widehat{\mathcal{M}}$ as a surrogate for samples from $\mathcal{M}$, leading to the objective:

$$\min_{\pi} \ \mathbb{D}(\rho_{\widehat{\mathcal{M}}}^{\pi}, \rho_{\mathcal{M}}^{E}), \tag{3}$$

which can serve as a good proxy to Eq. 1 as long as the model approximation is accurate. This intuition can be captured using the following lemma (see the appendix for proof).

**Lemma 1.** *(Simultaneous policy and model deviation) Suppose we have an $\alpha$-approximate dynamics model given by $\mathbb{D}_{TV}(\widehat{\mathcal{T}}(\boldsymbol{s}, \boldsymbol{a}), \mathcal{T}(\boldsymbol{s}, \boldsymbol{a})) \leq \alpha \ \forall (\boldsymbol{s}, \boldsymbol{a})$. Let $R_{\max} = \max_{(s,a)} \mathcal{R}(\boldsymbol{s}, \boldsymbol{a})$ be the maximum of the unknown reward in the MDP with unknown dynamics $\mathcal{T}$. For any policy $\pi$, we can bound the sub-optimality with respect to the expert policy $\pi^{E}$ as:*

$$\left| J(\pi^{E}, \mathcal{M}) - J(\pi, \mathcal{M}) \right| \leq \frac{R_{\max}}{1 - \gamma} \mathbb{D}_{TV}(\rho_{\widehat{\mathcal{M}}}^{\pi}, \rho_{\mathcal{M}}^{E}) + \frac{\alpha \cdot R_{\max}}{(1 - \gamma)^2}. \tag{4}$$

Thus, the divergence minimization in Eq. 3 serves as an approximate bound on the sub-optimality with a bias that is proportional to the model error. Thus, we ultimately propose to solve the following saddle point optimization problem:

$$\max_{\pi} \ \min_{D_{\psi}} \ \mathbb{E}_{(\boldsymbol{s}, \boldsymbol{a}) \sim \rho_{\mathcal{M}}^{E}} \left[ -\log D_{\psi}(\boldsymbol{s}, \boldsymbol{a}) \right] + \mathbb{E}_{(\boldsymbol{s}, \boldsymbol{a}) \sim \rho_{\widehat{\mathcal{M}}}^{\pi}} \left[ -\log \left( 1 - D_{\psi}(\boldsymbol{s}, \boldsymbol{a}) \right) \right], \tag{5}$$

which requires generating on-policy samples only from the learned model $\widehat{\mathcal{M}}$. We can interleave policy learning according to Eq. 5 with performing policy rollouts in the real environment to iteratively improve the model. Provided the policy is updated sufficiently slowly, Rajeswaran et al. [21] show that such interleaved policy and model learning corresponds to a stable and convergent algorithm, while being highly sample efficient.

## 3.2 Extension to POMDPs

In POMDPs, the underlying state is not directly observed, and thus cannot be directly used by the policy. In this case, we typically use the notion of *belief state*, which is defined to be the filtering distribution $P(\boldsymbol{s}_t | \boldsymbol{h}_t)$, where we denote history with $\boldsymbol{h}_t := (\boldsymbol{x}_{\leq t}, \boldsymbol{a}_{<t})$. By using the historical information, the belief state provides more information about the current state, and can enable the learning of better policies. However, learning and maintaining an explicit distribution over states can be difficult. Thus, we consider learning a latent representation of the history $\boldsymbol{z}_t = q(\boldsymbol{h}_t)$, so that $P(\boldsymbol{s}_t | \boldsymbol{h}_t) \approx P(\boldsymbol{s}_t | \boldsymbol{z}_t)$. To develop an algorithm for the POMDP setting, we first make the key observation that imitation learning in POMDPs can be reduced to divergence minimization in the latent belief state representation. To formalize this intuition, we introduce Theorem 1.

---

**Algorithm 1** V-MAIL: Variational Model-Based Adversarial Imitation Learning

---
1: **Require**: Expert demos $\mathcal{B}_E$, environment buffer $\mathcal{B}_\pi$.
2: Randomly initialize variational model $\{q_\theta, \widehat{\mathcal{T}}_\theta\}$, policy $\pi_\psi$ and discriminator $D_\psi$
3: **for** number of iterations **do**
4:    `// Environment Data Collection`
5:    **for** timestep $t = 1 : T$ **do**
6:        Estimate latent state from the belief distribution $z_t \sim q_\theta(\cdot|x_t, z_{t-1}, a_{t-1})$
7:        Sample action $a_t \sim \pi_\psi(a_t|z_t)$
8:        Step environment and get observation $x_{t+1}$
9:    Add data $\{x_{1:T}, a_{1:T-1}\}$ to policy replay buffer $\mathcal{B}_\pi$
10:    **for** number of training iterations **do**
11:        `// Dynamics Learning`
12:        Sample a batch of trajectories $\{x_{1:T}, a_{1:T-1}\}$ from the joint buffer $\mathcal{B}_E \cup \mathcal{B}_\pi$
13:        Optimize the variational model $\{q_\theta, \widehat{\mathcal{T}}_\theta\}$ using Equation 7
14:        `// Adversarial Policy Learning`
15:        Sample trajectories from expert buffer $\{x_{1:T}^E, a_{1:T-1}^E\} \sim \mathcal{B}_E$
16:        Infer expert latent states $z_{1:T}^E \sim q_\theta(\cdot|x_{1:T}^E, a_{1:T-1}^E)$ using the belief model $q_\theta$
17:        Generate latent rollouts $z_{1:H}^{\pi_\psi}$ using the policy $\pi_\psi$ from the forward model $\widehat{\mathcal{T}}_\theta$
18:        Update the discriminator $D_\psi$ with data $z_{1:T}^E, z_{1:H}^{\pi_\psi}$ using Equation 6
19:        Update the policy $\pi_\psi$ to improve the value function in Equation 8

---

**Theorem 1.** *(Divergence in latent space) Consider a POMDP $\mathcal{M}$, and let $z_t$ be a latent space representation of the history and belief state such that $P(s_t|x_{\leq t}, a_{< t}) = P(s_t|z_t)$. Let the policy class be such that $a_t \sim \pi(\cdot|z_t)$, so that $P(s_t|z_t, a_t) = P(s_t|z_t)$. Let $D_f$ be a generic $f-$divergence. Then the following inequalities hold:*

$$D_f(\rho_\mathcal{M}^\pi(x, a)||\rho_\mathcal{M}^E(x, a)) \leq D_f(\rho_\mathcal{M}^\pi(s, a)||\rho_\mathcal{M}^E(s, a)) \leq D_f(\rho_\mathcal{M}^\pi(z, a)||\rho_\mathcal{M}^E(z, a))$$

The condition $P(s_t|z_t, a_t) = P(s_t|z_t)$ essentially states that the actions of both the agent and the expert do not carry additional information about the state beyond what is available in the history. This will be true of all agents trained based on some representation of the history, and only excludes policies trained on ground truth states. Since we cannot hope to compete with policy classes that fundamentally have access to more information like the ground truth state, we believe this is a benign assumption. Theorem 1 suggests that the divergence of visitation distributions in the latent space represents an upper bound of the divergence in the state and observation spaces. This is particularly useful, since we do not have access to the ground-truth states of the POMDP and matching the expert marginal distribution in the high-dimensional observation space (such as images) could be difficult.

Furthermore, based on the results in Section 2.1, minimizing the state divergence results in minimizing a bound on policy sub-optimality as well. These results provide a direct way to extend the results from Section 3.1 to the POMDP setting. If we can learn an encoder $z_t = q(x_{\leq t}, a_{< t})$ that captures sufficient statistics of the history, and a latent state space dynamics model $z_{t+1} \sim \widehat{\mathcal{T}}(\cdot|z_t, a_t)$, then we can learn the policy by extending Eq. 5 to the induced MDP in the latent space as:

$$\max_\pi \min_{D_\psi} \mathbb{E}_{(z,a)\sim\rho_\mathcal{M}^E(z,a)}\left[-\log D_\psi(z, a)\right] + \mathbb{E}_{(z,a)\sim\rho_{\widehat{\mathcal{M}}}^\pi(z,a)}\left[-\log\left(1 - D_\psi(z, a)\right)\right]. \quad (6)$$

Once learned, the policy can be composed with the encoder for deployment in the POMDP. Similar approach was also taken by [22], however they only use the model for representation purposes in low-dimensional domains and do not carry out model-based training.

## 3.3 Practical Algorithm with Variational Models

The divergence bound of Theorem 1 allows us to develop a practical algorithm if we can learn a good belief state representation. Following prior work [23, 24, 25, 26, 27, 28] we optimize the ELBO:

$$\max_\theta \widehat{\mathbb{E}}_{q_\theta}\left[\sum_{t=1}^T \underbrace{\log\widehat{\mathcal{U}}_\theta(x_t|z_t)}_{\text{reconstruction}} - \underbrace{\mathbb{D}_{KL}(q_\theta(z_t|x_t, z_{t-1}, a_{t-1})||\widehat{\mathcal{T}}_\theta(z_t|z_{t-1}, a_{t-1}))}_{\text{forward model}}\right]. \quad (7)$$

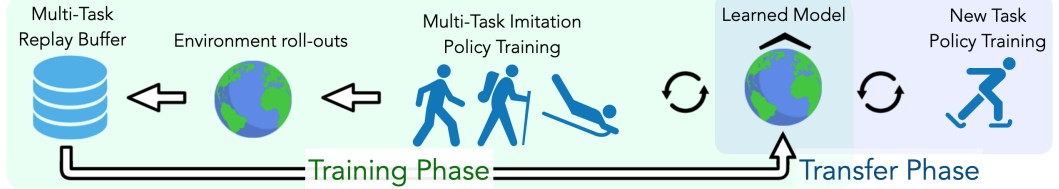

Figure 2: Illustration of our transfer learning approach. In the training phase, we learn a multiple tasks with a shared replay buffer and model. Subsequently, in the transfer and evaluation phase, the agent learns a new task using expert demonstrations and the learned model, without any additional interactions with the environment.

where $q_\theta$ is a state inference network, $\widehat{\mathcal{T}}_\theta$ is a latent dynamics model and $\widehat{\mathcal{U}}$ is an observation model. Here we jointly train a belief representation with network $q_\theta$ and a latent dynamics model $\widehat{\mathcal{T}}_\theta$. Given the learned latent model we can use any on-policy RL algorithm to train the policy using Eq. 6, however in our setup, the RL objective is a differentiable function of the policy, model, and discriminator parameters. We can then optimize the policy by directly back-propagating through $\widehat{\mathcal{T}}_\theta$ using the objective:

$$\max_{\pi_\psi} V^K_{\theta,\psi}(\boldsymbol{z}_t) = \max_{\pi_\psi} \mathbb{E}_{\pi_\psi, \widehat{\mathcal{T}}_\theta} \left[ \sum_{\tau=t}^{t+K-1} \gamma^{\tau-t} \log D_\psi(\boldsymbol{z}_\tau^{\pi_\psi}, \boldsymbol{a}_\tau^{\pi_\psi}) + \gamma^K V_\psi(\boldsymbol{z}_{t+K}^{\pi_\psi}) \right] \quad (8)$$

Finally, we train the discriminator $D_\psi$ using Eq. 5 with on-policy rollots from the model $\widehat{\mathcal{T}}$. Our full approach is outlined in Algorithm 1, for more details, see the appendix.

### 3.4 Zero-Shot Transfer to New Imitation Tasks

Our model-based approach is well suited to the problem of zero-shot transfer to new imitation learning tasks, i.e. transferring to a new task using a modest number of demonstrations and no additional samples collected in the environment.. In particular, we assume a set of source tasks $\{\mathcal{T}^i\}$, each with a buffer of expert demonstrations $\mathcal{B}_E^i$. Each source task corresponds to a different POMDP with different underlying rewards, but shared dynamics. During training, the agent can interact with each source environment and collect additional data. At test time, we're introduced with a new target task $\mathcal{T}$ with corresponding expert demonstrations $\mathcal{B}_E$ and the goal is to obtain a policy that achieves high reward without additional interaction with the environment.

Our proposed method is illustrated in Fig. 2. The key observation is that we can optimize Eq. 6 under our model and still obtain an upper bound on policy sub-optimality via Eq. 4. Furthermore, the sub-optimality is bound by the accuracy of our model over the marginal state-action distribution of the target task expert. Specifically, we first train on all of the source tasks using Algorithm 1, training a single shared variational model across the tasks. By fine-tuning that model on data that includes the target task expert demonstrations our hope is that we can get an accurate model and thus a high-quality policy. Similarly to Algorithm 1, we then train a discriminator and policy for the target task using only model rollouts. This approach is outlined in Algorithm 2.

---

**Algorithm 2** Zero-Shot Transfer with V-MAIL

1: **Require**: Expert demos $\mathcal{B}_E^i$ for each source task, expert demos $\mathcal{B}_E$ for target task
2: Randomly initialize policy $\pi_\psi$, and discriminator $D_\psi$
3: Train Alg 1 on source tasks, yielding shared model $\{q_\theta, \widehat{\mathcal{T}}_\theta\}$ and aggregated replay buffer $\mathcal{B}_\pi$
4: **for** number of training iterations **do**
5:      // Dynamics Fine-Tuning using Expert Trajectories
6:      Update the variational model $\{q_\theta, \widehat{\mathcal{T}}_\theta\}$ using Equation 7 with data from $\mathcal{B}_E \cup \mathcal{B}_\pi$
7:      // Adversarial Policy Learning
8:      Update discriminator $D_\psi$ and policy $\pi_\psi$ with Equations 6 and 8.

---

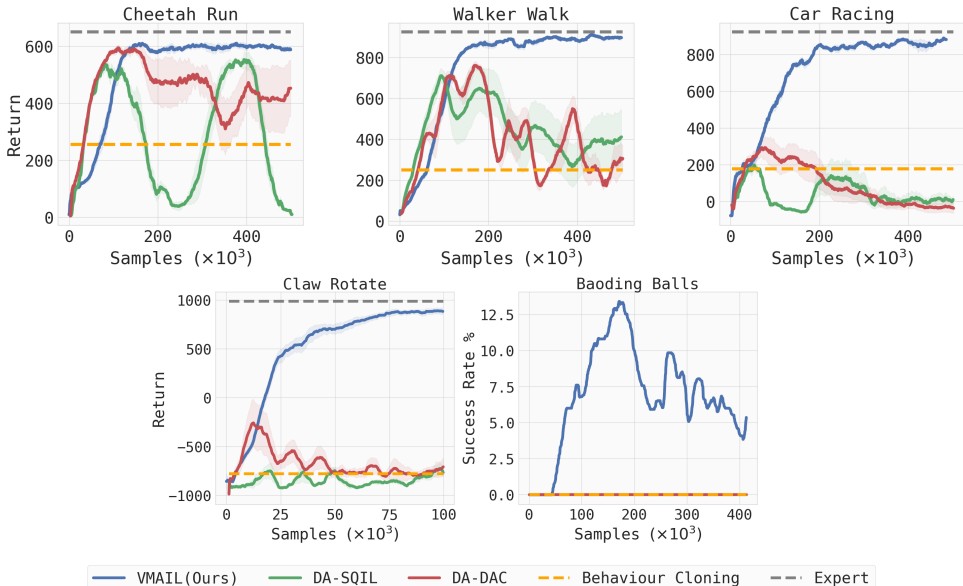

Figure 3: Learning curves showing ground truth reward versus number of environment steps for V-MAIL (ours), prior model-free imitation learning approaches, and behavior cloning on five visual imitation tasks. We find that V-MAIL consistently outperforms prior methods in terms of final performance and stability, particularly for the first four environments where V-MAIL reaches near-expert performance. In the most challenging visual Baoding Balls task, which is notably difficult even with ground-truth state, only V-MAIL is able to make some progress, but all methods struggle. Confidence intervals are shown with 1 SE over 6 runs.

## 4 Experiments

In our experiments, we aim to answer four questions: (1) can V-MAIL successfully solve environments with image observations, (2) how does V-MAIL compare to state-of-the-art model-free imitation approaches, (3) can V-MAIL solve realistic manipulation tasks and environments with complex physical interactions, and (4) can V-MAIL enable zero-shot transfer to new tasks? All experiments were carried out on a single Titan RTX GPU using an internal cluster for about 1000 GPU hours.

### 4.1 Single-Task Experiments

**Comparisons.** To answer question (2), we choose to compare V-MAIL to model-free adversarial and non-adversarial imitation learning methods. For the former, we choose DAC [18] as a representative approach, which we equip with DrQ data augmentation for greater performance on vision-based tasks. For the latter, we consider SQIL [29], also equipped with DrQ training. We refer to each approach with data augmentation as DA-DAC and DA-SQIL respectively. Both of these methods are off-policy algorithms, which we expect to be considerably more sample efficient than on-policy methods like GAIL [12] and AIRL [11]. For implementation details, see the appendix.

**Environments and Demonstration Data.** To answer the above questions, we consider the five visual control environments. These consist of two locomotion environments from the DeepMind Control Suite [30], the classic Car Racing environment from OpenAI Gym [31] and two dexterous manipulation tasks using the D'Claw [32] and Shadow Hand platforms. For full details on these environments and the expert data, see the appendix.

**Results.** Experiment results are shown in Figure 3. To answer questions (1) and (2), we compare V-MAIL to DA-SQIL and DA-DAC on the Cheetah and Walker tasks. We find that V-MAIL efficiently and reliably solves both tasks; in contrast, the model-free methods initially outperform V-MAIL, but their performance has high variance across random seeds and exhibits significant instability. Such stability issues have also been observed by Swamy et al. [33], which provides some theoretical explanation in the case of SQIL and the suggestion of early stopping as a mitigation technique. In the case of DAC, the reasons for instability are less clear. Motivated by instability we observed in the critic loss for DA-DAC, we experimented with a number of mitigation strategies in an attempt

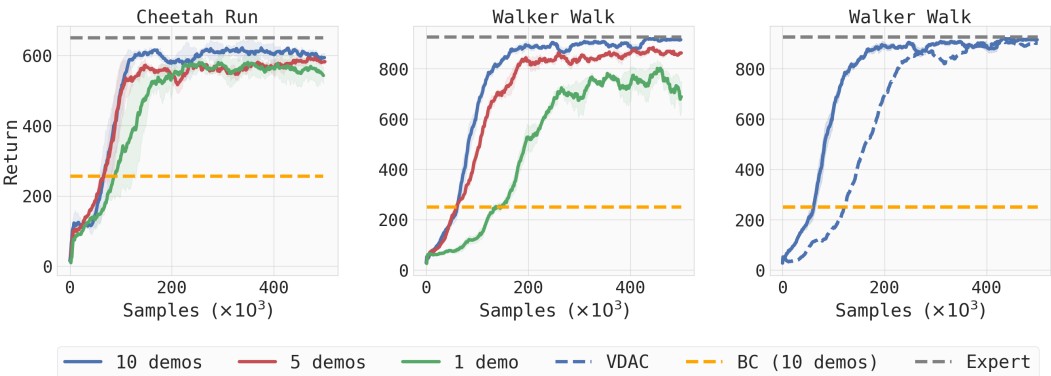

Figure 4: Ablation experiments for the VMAIL model. The left and middle graph show the efficacy of learning from different number of demonstrations where VMAIL outperforms baseline models even with a single demo. The final figure compares VMAIL to a model which used the variatonal model for representation purposes only (labeled VDAC). We see that VMAIL achieves 30% higher sample efficiency. Confidence intervals are shown with 1 SE over 3 runs.

to improve DA-DAC, including constraining the discriminator, varying the buffer and batch sizes, and separating the convolutional encoders of the discriminator and the actor/critic; however, these techniques didn't fully prevented the degradation in performance.

On the Car Racing environment, we find that DA-SQIL and DA-DAC can reach or outperform behavior cloning, but struggle to reach expert-level performance. In contrast, V-MAIL stably and reliably achieves near-expert performance in about 200k environment steps. Note that Reddy et al. [29] report expert-level performance on this task, but in an easier setting with double the number of expert demonstrations available (20 vs. 10). Given that tracks are randomly generated per episode demanding significant generalization, it is not surprising that the problem becomes considerably more difficult with only 10 demonstrations.

Finally, to answer question (3), we consider the D'Claw and Baoding Balls tasks. In the D'Claw environment, SQIL fails to make progress, while DA-DAC makes significant progress initially but quickly degrades. V-MAIL solves the task in less than 100k environment steps. In the most challenging visual Baoding Balls problem, involving a 26-dimensional control space, V-MAIL is the only algorithm to reach any success.

## 4.2 Ablation Experiments

Results for our ablation experiments are shown in Fig. 4.

**Number of Demonstrations.** In the first set of experiments we evaluate VMAIL's capability to learn from a limited number of demonstrations. We deploy our model on the two locomotion environments using 10, 5 and a single demonstration trajectory. VMAIL shows only minor deterioration in performance on the Cheetah Run environment, as well as on the Walker Walk environment when using only 5 demos, but it struggles to reach expert-level performance on the walker task when provided with a single trajectory. However, in all cases VMAIL outperforms the asymptotic results of our baselines, which use the full 10 demonstrations.

**Effect of Representation Learning.** We also evaluate the effect of representation learning on model performance. In this scenario we still train a variational model, but similar to [25] only use it for for representation purposes. We then train a standard Discriminator-Actor Critic model on top of the learned latent space, which we denote as Variational DAC or VDAC. Results are shown in the last graph of Fig. 4. VDAC exhibits more stable performance as compared to learning directly from pixels, however it is still 30% less sample efficient than VMAIL.

### 4.3 Transfer Experiments

**Transfer Scenarios.** To evaluate V-MAIL's ability to learn new imitation tasks in a zero-shot way (i.e. without any additional environment samples) we deploy Algorithm 2 on two domains: in a locomotion experiment we train on the Walker Stand and Walker Run (target speed greater than 8) tasks and and evaluate transfer to the Walker Walk (target speed between 2 and 4) task from the DeepMind Control suite. In a manipulation scenario, we use a set of custom D'Claw Screw tasks from the Robel suite [32]. We train our model on the 3-prong tasks with clockwise and counter-clockwise rotation, as well as the 4-prong task with counter-clockwise rotation and evaluate transfer to the 4-prong task with clockwise rotation.

**Comparisons.** We devise several points of comparison. First, we compare to directly applying the policy learned in the most related source task to the target task. This tests whether the target task demands qualitatively distinct behavior. Second, we compare to an offline version of DAC, augmented with the CQL approach [34], where samples collected from the source task are used to update the policy, with the target task demonstrations used to learn the reward. Finally, we also compare to behavior cloning on the target task demonstrations (without leveraging any source task data), and an oracle that performs V-MAIL on the target task directly.

**Results.** The results in Table 1. Policy transfer performs poorly, suggesting that the target task indeed requires qualitatively different behaviour from the few training tasks available. Further, behavior cloning on the target demonstrations is not sufficient to learn the task. Offline DAC also shows poor performance. Finally, we see that V-MAIL almost matches the performance of the agent explicitly trained on task, indicating the learned model and the algorithm for training within that model can be used not just for efficient visual imitation learning, but also for zero-shot transfer to new tasks.

| Method | Walker Walk | Claw Rotate |
|---|---|---|
| Offline DAC | 8.8% | -0.7% |
| Behavior cloning | 26.8% | 8.3% |
| Policy transfer | 21.3% | 5.6% |
| V-MAIL (ours) | **92.7%** | **97.9%** |
| Target task IL (oracle) | 98.2% | 102.3% |

Table 1: Zero-shot transfer performance to a new imitation learning task as percent of expert return. Each method is provided with 10 demonstrations of the target task, and zero additional environment samples. V-MAIL can solve the target tasks within its learned model without any additional samples, while model-free transfer learning approaches fail.

## 5 Related Work

**Imitation Learning.** Recent model-free imitation learning can be categorized as either adversarial or non-adversarial. Adversarial methods inspired by GANs [15] train an explicit classifier between expert and policy behaviour and optimize the agent in a two-player minimax game. GAIL [12] and AIRL [11] are two such algorithms; however they often have poor sample efficiency due to the requirement of on-policy rollouts in the environment. To address sample efficiency issues, off-policy variants such as DAC [18] and SAM [19] have been developed, however they suffer from an objective mismatch when using off-policy data [35], often resulting in learning instability [36].

An alternate line of research attempts to forego adversarial training: SQIL [29] frames the problem as regularized behaviour cloning and trains an off-policy algorithm with rewards of 1 for expert trajectories and 0 for policy ones. RCE [37] uses a very similar approach, but derives it through a different objective. ValueDICE [35] uses the same key result for iterative distribution matching as RCE to obtain an off-policy distribution matching algorithm. In Swamy et al. [33] the authors derive distribution matching as a bound on policy under-performance, similar to our analysis in Section 3.1 and propose a practical non-adversarial algorithm AdVIL, however in reported experiments it does not outperform behaviour cloning. In Berseth et al. [38] the authors consider a variational encoder, which embeds expert video demonstrations in a latent space, and optimize a mis-match objective using on-policy model-free training.

A few papers have considered model-based imitation learning as well: Baram et al. [39] is an adversarial algorithm conceptually similar to our approach, but only focuses on low-dimensional state-based tasks and train the discriminator using off-policy replay buffer, which does not allow it to generalize to new tasks. Related to our method is Finn et al. [10] which uses a similar reward learning in combination with a locally linear dynamics model, which leads to trajectory-centric algorithms that cannot transfer the model to new tasks. Das et al. [40] considers a similar setting for inverse

RL using a simplified parameterization of the cost function. In this work we develop end-to-end model for adversarial imitation learning in high-dimensional POMDPs and generalization to novel tasks without hand-designed features. A related line of research is learning from observations alone (without access to expert actions). The BCO approach Torabi et al. [41] learns an inverse dynamics model to infer and copy expert actions, however it still suffers from the drawbacks of behaviour cloning. In a concurrent work [42] explore similar objective to [43] and VMAIL, using only states for training the discriminator objective, however they only focus on low dimensional domains. With a particular variational model specification VMAIL can be also be extended to work only with expert state observations, which we leave for further work.

Finally, related to zero-shot imitation learning, concurrent work by Chang et al. [43] extends a model-based offline RL algorithm [44, 45] to imitation learning. The focus in their work is primarily theoretical analysis, with empirical results in tasks with compact state representations. In contrast, our work aims to develop a stable and efficient imitation learning algorithm that can handle high-dimensional observation spaces (like visual inputs), in both online and offline learning settings.

**Reinforcement Learning From Images with Variational Models.** Reinforcement learning from images is an inherently difficult task, since the agent needs to learn meaningful visual representations to support policy learning. A recent line of research [26, 27, 25, 28, 46] train a variational model of the image-based environment as an auxiliary task, either for representation learning only [26, 25] or for additionally generating on-policy data by rolling out the model [28]. Our method builds upon these ideas, but unlike these prior works, considers the problem of learning from visual demonstrations without access to rewards.

## 6 Conclusion

In this work we presented V-MAIL, a model-based imitation learning algorithm that works from high-dimensional image observations. V-MAIL learns a model of the environment, which serves as a strong self-supervision signal for visual representation learning and mitigates distribution shift by enabling synthetic on-policy rollouts using the model. Through experiments, we find that V-MAIL achieves better asymptotic performance, is more stable, and matches the sample efficiency of prior model-free approaches. By effectively re-using the learned model, V-MAIL is also successful in zero-shot imitation learning, capable of learning new tasks using a small number of demonstrations, without any additional interactions with the environment.

**Limitations.** VMAIL uses a low-dimensional dynamics model, explicitly trained with image prediction. This might make our method vulnerable to adversarial visual perturbations. Although successful in domains with complex dynamics, our approach relies on variational models with compact, single-level, latent state spaces. It is possible that such a model class may not have sufficient capacity to represent complex scenes with multiple cluttered objects or deformable objects. A different selection of training objectives or further developments and improvements in variational generative modeling can potentially address both of these limitations.

**Future Work.** We believe this work opens the door for multiple potential developments. One direction is to train our procedure using only expert observations without access to expert actions, which is an even more realistic scenario. This setup is quite difficult for model-free approaches, since expert actions usually serve as a strong supervision. Another direction is to use on-policy model based rollouts to efficiently train other algorithms that inherently require on-policy data, such as multi-modal imitation [47, 48]. We showed the transfer capabilities of our algorithm to new tasks in a zero-shot imitation learning formulation. However, V-MAIL can in principle utilize any previously collected data for model-training, enabling potential applications in offline imitation learning in conjunction with offline RL algorithms like Rafailov et al. [46].

## Acknowledgements

This work was supported in part byt ONR grants N00014-20-1-2675 and N00014-21-1-2685 as well as Intel Corporation. Chelsea Finn is a CIFAR Fellow in the Learning in Machines and Brains program. Part of this work was completed when Aravind Rajeswaran was at the University of Washington, where he was supported through the J.P. Morgan PhD Fellowship in AI (2020-21).

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
