## A Theoretical Results

We base our approach on the following theoretical results from the paper.

**Theorem 1 Restated.** *Consider a POMDP $\mathcal{M}$, and let $z_t$ be a latent space representation of the history and belief state such that $P(s_t|x_{<t}, a_{<t}) = P(s_t|z_t)$. Let the policy class be such that $a_t \sim \pi(\cdot|z_t)$, so that $P(s_t|z_t, a_t) = P(s_t|z_t)$. Let $D_f$ be a generic $f-$divergence. Then the following inequalities hold:*

$$D_f(\rho_{\mathcal{M}}^{\pi}(x,a)||\rho_{\mathcal{M}}^{E}(x,a)) \leq D_f(\rho_{\mathcal{M}}^{\pi}(s,a)||\rho_{\mathcal{M}}^{E}(s,a)) \leq D_f(\rho_{\mathcal{M}}^{\pi}(z,a)||\rho_{\mathcal{M}}^{E}(z,a))$$

*Proof.* The condition $P(s_t|z_t, a_t) = P(s_t|z_t)$ essentially states that the actions of both the agent and the expert do not carry additional information about the state beyond what is available in the history. This will be true of all agents trained based on some representation of the history or just the current observation, and only excludes policies trained on ground truth states. Since we cannot hope to compete with policy classes that fundamentally have access to more information like the ground truth state, we believe this is a benign assumption. With these assumptions, the proof is straightforward application of the data-processing inequality [49].

$$D_f(\rho_{\mathcal{M}}^{\pi}(z,a)||\rho_{\mathcal{M}}^{E}(z,a)) = \mathbb{E}_{z,a\sim\rho_{\mathcal{M}}^{E}(z,a)}\left[f\left(\frac{\rho_{\mathcal{M}}^{\pi}(z,a)}{\rho_{\mathcal{M}}^{E}(z,a)}\right)\right] \tag{9}$$

$$= \mathbb{E}_{z,a\sim\rho_{\mathcal{M}}^{E}(z,a)}\mathbb{E}_{s\sim P(s|z)}\left[f\left(\frac{\rho_{\mathcal{M}}^{\pi}(z,a)}{\rho_{\mathcal{M}}^{E}(z,a)}\frac{P(s|z)}{P(s|z)}\right)\right] \tag{10}$$

$$= \mathbb{E}_{z,s,a\sim\rho_{\mathcal{M}}^{E}(z,s,a)}\left[f\left(\frac{\rho_{\mathcal{M}}^{\pi}(z,s,a)}{\rho_{\mathcal{M}}^{E}(z,s,a)}\right)\right] \tag{11}$$

$$= \mathbb{E}_{s,a\sim\rho_{\mathcal{M}}^{E}(s,a)}\left[\mathbb{E}_{z\sim\rho_{\mathcal{M}}^{E}(z|s,a)}f\left(\frac{\rho_{\mathcal{M}}^{\pi}(z,s,a)}{\rho_{\mathcal{M}}^{E}(z,s,a)}\right)\right] \tag{12}$$

$$\geq \mathbb{E}_{s,a\sim\rho_{\mathcal{M}}^{E}(s,a)}\left[f\left(\mathbb{E}_{z\sim\rho_{\mathcal{M}}^{E}(z|s,a)}\frac{\rho_{\mathcal{M}}^{\pi}(z,s,a)}{\rho_{\mathcal{M}}^{E}(z,s,a)}\right)\right] \tag{13}$$

$$= \mathbb{E}_{s,a\sim\rho_{\mathcal{M}}^{E}(s,a)}\left[f\left(\mathbb{E}_{z\sim\rho_{\mathcal{M}}^{E}(z|s,a)}\frac{\rho_{\mathcal{M}}^{\pi}(s,a)\rho_{\mathcal{M}}^{\pi}(z|s,a)}{\rho_{\mathcal{M}}^{E}(s,a)\rho_{\mathcal{M}}^{E}(z|s,a)}\right)\right] \tag{14}$$

$$= \mathbb{E}_{s,a\sim\rho_{\mathcal{M}}^{E}(s,a)}\left[f\left(\mathbb{E}_{z\sim\rho_{\mathcal{M}}^{\pi}(z|s,a)}\frac{\rho_{\mathcal{M}}^{\pi}(s,a)}{\rho_{\mathcal{M}}^{E}(s,a)}\right)\right] \tag{15}$$

$$= \mathbb{E}_{s,a\sim\rho_{\mathcal{M}}^{E}(s,a)}\left[f\left(\frac{\rho_{\mathcal{M}}^{\pi}(s,a)}{\rho_{\mathcal{M}}^{E}(s,a)}\right)\right] \tag{16}$$

$$= D_f(\rho_{\mathcal{M}}^{\pi}(s,a)||\rho_{\mathcal{M}}^{E}(s,a)) \tag{17}$$

The first two equalities (9-10) follow from the fact that $\rho_{\mathcal{M}}^{\pi}(s|z,a) = P(s|z) = \rho_{\mathcal{M}}^{E}(s|z,a)$ from the assumptions of the Theorem. The inequality (13) is a direct application of Jensen's inequality and the definition of an $f-$divergence. The other part of the main result follows the same reasoning, considering the observation model $\mathcal{U}(x|s)$, rather than the belief distribution $P(s|z)$. We also refer readers to Gangwani et al. [22] for similar derivations and analysis. $\square$

**Lemma 1 Restated.** *Suppose we have an $\alpha$-approximate dynamics model given by $\mathbb{D}_{TV}(\widehat{\mathcal{T}}(s,a), \mathcal{T}(s,a)) \leq \alpha \ \forall(s,a)$. Let $R_{\max} = \max_{(s,a)} \mathcal{R}(s,a)$ be the maximum of the unknown reward in the MDP with unknown dynamics $\mathcal{T}$. For any policy $\pi$, we can bound the sub-optimality with respect to the expert policy $\pi^E$ as:*

$$\left|J(\pi^E, \mathcal{M}) - J(\pi, \mathcal{M})\right| \leq \frac{R_{\max}}{1-\gamma}\mathbb{D}_{TV}(\rho_{\widehat{\mathcal{M}}}^{\pi}, \rho_{\mathcal{M}}^{E}) + \frac{\alpha \cdot R_{\max}}{(1-\gamma)^2}.$$

*Proof.* The proof is a simple application of the triangle inequality on $\mathbb{D}_{TV}$. In particular we have:

$$\left| J(\pi^E, \mathcal{M}) - J(\pi, \mathcal{M}) \right| \leq \frac{R_{\max}}{1 - \gamma} \, \mathbb{D}_{TV}(\rho_\mathcal{M}^\pi, \rho_\mathcal{M}^E) \tag{18}$$

$$\leq \frac{R_{\max}}{1 - \gamma} \, \left( \mathbb{D}_{TV}(\rho_{\widehat{\mathcal{M}}}^\pi, \rho_\mathcal{M}^E) + \mathbb{D}_{TV}(\rho_{\widehat{\mathcal{M}}}^\pi, \rho_\mathcal{M}^\pi) \right) \tag{19}$$

$$\leq \frac{R_{\max}}{1 - \gamma} \, \mathbb{D}_{TV}(\rho_{\widehat{\mathcal{M}}}^\pi, \rho_\mathcal{M}^E) + \frac{\alpha \cdot R_{\max}}{(1 - \gamma)^2}. \tag{20}$$

Eq. (18) is directly from the definition, $(1 - \gamma) \cdot J(\pi, \mathcal{M}) = \mathbb{E}_{(\boldsymbol{s}, \boldsymbol{a}) \sim \rho_\mathcal{M}^\pi} \left[ r(\boldsymbol{s}, \boldsymbol{a}) \right]$. Eq. (19) is based on triangle inequality by decomposing with $\rho_{\widehat{\mathcal{M}}}^\pi$ as an intermediate variable. The final inequality in Eq. (20) is a direct consequence of error amplification lemma from Rajeswaran et al. [21]. $\square$

# B   Practical Algorithm with Variational Models

The divergence bound of Theorem 1 allows us to develop a practical algorithm if we can learn a good belief state representation. Towards that end we turn to the theory of deep Bayesian filters [50] and begin with the likelihood:

$$\log P(\boldsymbol{x}_{1:T} | \boldsymbol{a}_{1:T}) = \log \int \prod_{t=1}^{T} \mathcal{U}(\boldsymbol{x}_t | \boldsymbol{s}_t) \mathcal{T}(\boldsymbol{s}_t | \boldsymbol{a}_{t-1}, \boldsymbol{s}_{t-1}) d\boldsymbol{s}_{1:T}$$

We can introduce the belief distribution $q(\boldsymbol{z}_{1:T} | \boldsymbol{x}_{1:T}, \boldsymbol{a}_{1:T-1}) = \prod_{t=1}^{T} q(\boldsymbol{z}_t | \boldsymbol{x}_t, \boldsymbol{z}_{t-1}, \boldsymbol{a}_{t-1})$, which considers only model classes that satisfy the the sufficient statistics requirement. Using the introduced belief distribution as the variational distribution, we derive the evidence lower bound (ELBO) [51, 52]:

$$\log P(\boldsymbol{x}_{1:T} | \boldsymbol{a}_{1:T}) = \log \int \prod_{t=1}^{T} \mathcal{U}(\boldsymbol{x}_t | \boldsymbol{z}_t) \mathcal{T}(\boldsymbol{z}_t | \boldsymbol{a}_{t-1}, \boldsymbol{z}_{t-1}) \frac{q(\boldsymbol{z}_t | \boldsymbol{x}_t, \boldsymbol{z}_{t-1}, \boldsymbol{a}_{t-1})}{q(\boldsymbol{z}_t | \boldsymbol{x}_t, \boldsymbol{z}_{t-1}, \boldsymbol{a}_{t-1})} d\boldsymbol{z}_{1:T} =$$

$$\log \int q(\boldsymbol{z}_{1:T} | \boldsymbol{x}_{1:T}, \boldsymbol{a}_{1:T-1}) \prod_{t=1}^{T} \mathcal{U}(\boldsymbol{x}_t | \boldsymbol{z}_t) \frac{\mathcal{T}(\boldsymbol{z}_t | \boldsymbol{a}_{t-1}, \boldsymbol{z}_{t-1})}{q(\boldsymbol{z}_t | \boldsymbol{x}_t, \boldsymbol{z}_{t-1}, \boldsymbol{a}_{t-1})} d\boldsymbol{z}_{1:T} =$$

$$\log \mathbb{E}_{q(\boldsymbol{z}_{1:T} | \boldsymbol{x}_{1:T}, \boldsymbol{a}_{1:T-1})} \left[ \prod_{t=1}^{T} \mathcal{U}(\boldsymbol{x}_t | \boldsymbol{z}_t) \frac{\mathcal{T}(\boldsymbol{z}_t | \boldsymbol{a}_{t-1}, \boldsymbol{z}_{t-1})}{q(\boldsymbol{z}_t | \boldsymbol{x}_t, \boldsymbol{z}_{t-1}, \boldsymbol{a}_{t-1})} \right] \geq$$

$$\mathbb{E}_{q(\boldsymbol{z}_{1:T} | \boldsymbol{x}_{1:T}, \boldsymbol{a}_{1:T-1})} \left[ \log \prod_{t=1}^{T} \mathcal{U}(\boldsymbol{x}_t | \boldsymbol{z}_t) \frac{\mathcal{T}(\boldsymbol{z}_t | \boldsymbol{a}_{t-1}, \boldsymbol{z}_{t-1})}{q(\boldsymbol{z}_t | \boldsymbol{x}_t, \boldsymbol{z}_{t-1}, \boldsymbol{a}_{t-1})} \right] =$$

$$\mathbb{E}_{q(\boldsymbol{z}_{1:T} | \boldsymbol{x}_{1:T}, \boldsymbol{a}_{1:T-1})} \left[ \sum_{t=1}^{T} \log \mathcal{U}(\boldsymbol{x}_t | \boldsymbol{z}_t) \right] + \mathbb{E}_{q(\boldsymbol{z}_{1:T} | \boldsymbol{x}_{1:T}, \boldsymbol{a}_{1:T-1})} \left[ \sum_{t=1}^{T} \frac{\mathcal{T}(\boldsymbol{z}_t | \boldsymbol{a}_{t-1}, \boldsymbol{z}_{t-1})}{q(\boldsymbol{z}_t | \boldsymbol{x}_t, \boldsymbol{z}_{t-1}, \boldsymbol{a}_{t-1})} \right] =$$

$$\mathbb{E}_{q(\boldsymbol{z}_{1:T} | \boldsymbol{x}_{1:T}, \boldsymbol{a}_{1:T-1})} \left[ \sum_{t=1}^{T} \log \mathcal{U}(\boldsymbol{x}_t | \boldsymbol{z}_t) \right] +$$

$$\sum_{t=1}^{T} \mathbb{E}_{q(\boldsymbol{z}_{1:t-1} | \boldsymbol{x}_{1:t-1}, \boldsymbol{a}_{1:t-2})} \left[ \mathbb{E}_{q(\boldsymbol{z}_t | \boldsymbol{x}_t, \boldsymbol{z}_{t-1}, \boldsymbol{a}_{t-1})} \frac{\mathcal{T}(\boldsymbol{z}_t | \boldsymbol{a}_{t-1}, \boldsymbol{z}_{t-1})}{q(\boldsymbol{z}_t | \boldsymbol{x}_t, \boldsymbol{z}_{t-1}, \boldsymbol{a}_{t-1})} \right] =$$

$$\mathbb{E}_{q(\boldsymbol{z}_{1:T}|\boldsymbol{x}_{1:T},\boldsymbol{a}_{1:T-1})}\left[\sum_{t=1}^{T}\log\mathcal{U}(\boldsymbol{x}_t|\boldsymbol{z}_t)\right]+$$

$$\sum_{t=1}^{T}\mathbb{E}_{q(\boldsymbol{z}_{1:t-1}|\boldsymbol{x}_{1:t-1},\boldsymbol{a}_{1:t-2})}[\mathbb{D}_{KL}(q(\boldsymbol{z}_t|\boldsymbol{x}_t,\boldsymbol{z}_{t-1},\boldsymbol{a}_{t-1})||\mathcal{T}(\boldsymbol{z}_t|\boldsymbol{z}_{t-1},\boldsymbol{a}_{t-1}))]$$

We represent $q, \mathcal{T}$ and $\mathcal{U}$ as neural networks with parameters $\theta$. Following standard deep variational inference techniques Kingma and Welling [52] all distributions are represented as diagonal Gaussians parameterized by neural networks. To estimate the expectation, we can use sequential sampling from the belief distribution $\boldsymbol{z}_t \sim q_\theta(\cdot|\boldsymbol{x}_t,\boldsymbol{z}_{t-1},\boldsymbol{a}_{t-1}), t = 1 : T$ and the reparameterization trick [52]. This leads to the empirical model loss:

$$\max_\theta \widehat{\mathbb{E}}_{q_\theta}\Big[\sum_{t=1}^{T}\underbrace{\log\widehat{\mathcal{U}}_\theta(\boldsymbol{x}_t|\boldsymbol{z}_t)}_{\text{reconstruction}}-\underbrace{\mathbb{D}_{KL}(q_\theta(\boldsymbol{z}_t|\boldsymbol{x}_t,\boldsymbol{z}_{t-1},\boldsymbol{a}_{t-1})||\widehat{\mathcal{T}}_\theta(\boldsymbol{z}_t|\boldsymbol{z}_{t-1},\boldsymbol{a}_{t-1}))}_{\text{forward model}}\Big]. \qquad (21)$$

That is, we jointly train a belief representation $q_\theta$ and the Markovian dynamics model $\widehat{\mathcal{T}}_\theta$, which allows us to optimize Eq. 5 in our learned belief space. A number of recent works have considered similar models [23, 24, 25, 26, 27, 28]. We base our network architectural choice on the recurrent state space model [27, 28], as it has shown strong performance in RL tasks from images.

Once we learn a low-dimensional model, any on-policy RL algorithm can be used to train the policy using Eq. 6. In our setup, the RL objective is a differentiable function of the policy, model, and discriminator parameters. Based on this, we setup a $K$ step value expansion objective [53, 54] given below, and use it for policy learning.

$$\max_{\pi_\psi} V_{\theta,\psi}^K(\boldsymbol{z}_t) = \max_{\pi_\psi}\mathbb{E}_{\pi_\psi,\widehat{\mathcal{T}}_\theta}\left[\sum_{\tau=t}^{t+K-1}\gamma^{\tau-t}\log D_\psi(\boldsymbol{z}_\tau^{\pi_\psi},\boldsymbol{a}_\tau^{\pi_\psi}) + \gamma^K V_\psi(\boldsymbol{z}_{t+K}^{\pi_\psi})\right] \qquad (22)$$

where the expectaion is taken over the sampling distributions $\boldsymbol{a}_t^{\pi_\psi} \sim \pi_\psi(\boldsymbol{a}|\boldsymbol{z}_t^{\pi_\psi})$ and $\boldsymbol{z}_{t+1}^{\pi_\psi} \sim \widehat{\mathcal{T}}_\theta(\boldsymbol{z}|\boldsymbol{z}_t^{\pi_\psi},\boldsymbol{a}_t^{\pi_\psi})$. This allows us to optimize the policy $\pi_\psi$ by directly differenting the above objective through the learned model dynamics. The value function is fitted using the bootstrapped estimate in Eq. 22.

Finally, we train the discriminator $D_\psi$ in the learned latent space using Eq. 5 with on-policy rollots from the model $\widehat{\mathcal{T}}$:

$$\min_{D_\psi}\mathbb{E}_{(\boldsymbol{z},\boldsymbol{a})\sim q_\theta(\rho_{\mathcal{M}}^E)}\left[-\log D_\psi(\boldsymbol{z}^E,\boldsymbol{a}^E)\right] + \mathbb{E}_{\boldsymbol{z}^{\pi_\psi},\boldsymbol{a}^{\pi_\psi}\sim\pi_\psi,\widehat{\mathcal{T}}_\theta}\left[-\log\left(1-D_\psi(\boldsymbol{z}^{\pi_\psi},\boldsymbol{a}^{\pi_\psi})\right)\right] \quad (23)$$

As outlined in Algorithm 1 we iteratively optimize the model, actor-critic and discriminator using Eq. 21, 22, 23.

## C  Practical Off-Policy Imitation Learning Algorithms

Training reinforcement learning policies from images is challenging using environment rewards, but even more so in the case of adversarial imitation learning. We explicitly choose to benchmark our method against SQIL and DAC, which use sample efficient off-policy training. In addition we can augment these approaches with state of the art method DrQ [55], which has shown up to two orders of magnitude improvement in sample efficiency when training policies from raw pixels. The key of the DrQ approach is to introduce a family of image-augmentation functions $f(\boldsymbol{s}, v)$, where $\boldsymbol{s}$ is an environment state (a set of stacked images) and $v$ are augmentation parameters, from a fixed set of transformations. Given a batch of transition tuples $(\boldsymbol{s}_i, \boldsymbol{a}_i, \boldsymbol{s}_i', \boldsymbol{r}_i)$ the standard Q-learning procedure is augmented as follows: the target values for the Bellman backups are computed as:

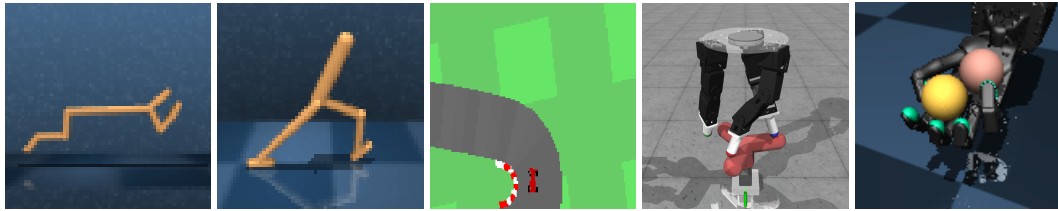

Figure 5: Illustration of the environments used in our experiments: Cheetah, Walker, Car Racing, D'Claw, and Baoding Balls. In all environments, the agent has access only to the RGB image frames as observations, except with additional access to proprioception in the Baoding Balls environment.

$$\boldsymbol{y}_i = \boldsymbol{r}_i + \gamma \frac{1}{K} \sum_{k=1}^{K} Q_\theta^{target}(f(\boldsymbol{s}_i', v_{i,k}'), \boldsymbol{a}_{i,k}') \text{ where } \boldsymbol{a}_{i,k}' \sim \pi(\cdot | f(\boldsymbol{s}_i', v_{i,k}')) \quad (24)$$

while the Q-function is updated by:

$$\theta \leftarrow \theta - \lambda \nabla_\theta \frac{1}{NM} \sum_{i=1,m=1}^{N,M} (Q_\theta(f(\boldsymbol{s}_i, v_{i,m}), \boldsymbol{a}_i) - \boldsymbol{y}_i)^2 \quad (25)$$

We can directly adapt SQIL to this setup, by using stationary rewards for the expert and policy replay buffers. For DAC, we train an additional discriminator $D_\psi$ minimizing the objective:

$$\mathbb{E}_{\boldsymbol{s},\boldsymbol{a} \sim \mathcal{B}^E} \left[ \frac{1}{K} \sum_{k=1}^{K} -\log D_\psi(f(\boldsymbol{s}, v_k), \boldsymbol{a}) \right] + \mathbb{E}_{\boldsymbol{s},\boldsymbol{a} \sim \mathcal{B}^\pi} \left[ \frac{1}{K} \sum_{k=1}^{K} -\log(1 - D_\psi(f(\boldsymbol{s}, v_k), \boldsymbol{a})) \right] \quad (26)$$

and then train the actor-critic algorithm with a modified version of Eq. 25:

$$\boldsymbol{y}_i = \frac{1}{K} \sum_{k=1}^{K} \log D_\psi(f(\boldsymbol{s}_i, v_{i,k}), \boldsymbol{a}_i) + \gamma Q_\theta^{target}(f(\boldsymbol{s}_i', v_{i,k}'), \boldsymbol{a}_{i,k}') \quad (27)$$

In our implementation the discriminator, critic and policy share the same convolutional encoder, which is trained using the discriminator and critic loss only. During training of this baseline, we noticed that periods of poor performance coincide with instability in the critic loss, rather than the discriminator. We hypothesise that this is potentially caused by issues with value function bootstrapping with non-stationary rewards or a mis-matched entropy objective from the soft actor-critic objective. We experimented with a number of mitigation strategies in an attempt to improve performance of this baseline, including constraining the discriminator, different regularization techniques, varying the buffer and batch sizes and separating the convolutional encoders of the discriminator and the actor/critic; however, these approaches didn't fully prevented the degradation in performance.

## D   Environments and Demonstration Data.

We consider the five visual control environments illustrated in Figure 5. The Cheetah Run and Walker Walk are standard tasks from the DeepMind Control Suite [30], with $64 \times 64$ pixel observations. Following SQIL [29] we also consider the Car Racing environment from OpenAI Gym [31]. We slightly modify the observation space, by removing the bottom part of the image, which contains episode and reward statistics and reshape the remaining image into $128 \times 128$ pixels. In addition, we benchmark our method on a custom D'Claw environment from the Robel suite [32]. The goal of the environment is to rotate the valve as fast as possible. We only use $128 \times 128$ image observations without proprioception, which makes the task challenging due to a complex action and contact dynamics, as well as occlusions from the robot fingers. Our final environment is the Baoding balls

task from Nagabandi et al. [56]. This is an extremely challenging task for policy learning, even in the state-based case. In addition to $128 \times 128$ images, we also include proprioception information from the robot hand in the observation space. This is not unrealistic since the real ShadowHand robot platform can provide such information.

In the main set of experiments, all methods receive access to 10 expert demonstrations, with the exception of the Baoding environment, which uses 25 demonstrations. The demonstrations for the DeepMind Control and D'Claw tasks are generated using a policy trained with SAC [57], the expert data for the Car Racing environment is generated using Dreamer [28], and the demonstrations for the Baoding task are generated using the PDDM framework [56] from low-dimensional states.

We use three Walker-based locomotion environments from the DeepMind Control Suite [30] for our transfer experiments. We train VMAIL on the Walker Stand and Walker Run tasks. The objective of the Walker Stand task is to maintain an upright position without moving, while the objective of the Run task is to maintain velocity greater than 8. We evaluate the transfer capabilities of VMAIL on Walker Walk, which aims to maintain velocity within the 2-4 range. We should note that these are qualitatively different behaviours. In a manipulation scenario, we use a set of custom D'Claw Screw tasks from the Robel suite Ahn et al. [32]. The goal of all tasks is to rotate the valve as fast as possible in a particular direction. We train our model on tasks with a 3-prong valve with clockwise and counter-clockwise rotation, as well as a task with a 4-prong valve with counter-clockwise rotation. We then evaluate the transfer to the task with a 4-prong valve with clockwise rotation. In all 4 environments we control the same 3-fingered D'Claw robot, but the shape of the valve object and the rotation direction vary.