# OpenReview forum: "Visual Adversarial Imitation Learning using Variational Models"
_NeurIPS.cc/2021/Conference — NeurIPS 2021 Poster_

### Official Review · Reviewer_LYjg · 2021-07-16

**Rating:** 6
**Confidence:** 4

**Summary:**

This paper introduces Variational model-based adversarial imitation learning (V-MAIL). V-MAIL considers a model-based setup for adversarial imitation learning: a latent dynamic model is learned and used to generate on-policy trajectories for training the RL policy generally used in the GAIL setup. It manages to achieve state-of-the-art performance on multiple tasks from visual observations, including MuJoCo, classic car racing, and two other robot manipulation tasks.


**Limitations And Societal Impact:**

I have certain concerns about the paper.

1/ I think the contribution of the paper is a bit limited. V-MAIL combines several existing ideas, e.g., latent imagination, GAIL, variational state space model, etc., and achieves a good performance. However, how these components affect each other and how they contribute to the final performance are not clear. An ablation study is also missing from the paper. In this case, it would be hard to get inspiration from reading it.

2/ Although the paper has provided theoretical analysis about the model-based adversarial imitation learning (sect. 4.1 and sect. 4.2), they are disconnected from the practical implementations (sect. 4.3). In particular, Theorem 1 shows that the divergence of the visitation distributions in a MDP can be upper bounded by a divergence of the visitation distributions in a POMDP. However, in the practical implementation, a variational state space model captures the belief, rather than the visitation distribution of the belief. In addition, it feels difficult to compute the visitation frequency of a belief, whose size is exponential to the size of the history and the state space. I believe the proposed algorithm indeed has its merit, but I don’t think Theorem 1 provides a correct justification of the optimization objective used in this paper.

3/ I feel the author should be careful when making certain claims. For example, from line 39 to line 48, the authors are analyzing the limitations of the existing IRL methods and adversarial imitation learning methods. “These approaches explicitly train a GAN-based classifier [17] to distinguish the visitation distribution of the agent from the expert, and use it as a reward signal for training the agent with RL….” However, not all IRL methods are adversarial imitation learning. In fact, most of them don’t train a GAN-based classifier and do RL afterwards. Instead, a lot of them recover the reward and do planning instead.

4/ The authors claimed that V-MAIL achieves zero-shot transfer to novel environments. However, the policy is fine-tuned with additional expert demonstrations, as shown in Alg. 2. Why is this zero-shot? In addition, I guess the transferability might be limited by the real difficulty of the source task / target task. Walker-run is clearly harder than walker-walk, so the policy transfer here is possible. Also, for the manipulation scenario, 3-prong task with both clockwise / counter clockwise rotations, together with one 4-prong task, actually provides sufficient information about the target task. I guess it might be difficult to transfer a policy from simpler tasks to more complex tasks. This has to be made clear in the paper. Otherwise, it is quite misleading.


**Main Review:**

Overall the paper is well-written and easy to follow. Combining the model-based learning with GAIL-like imitation learning has indeed given extra benefit: it works like a regulizer for the policy, and provides nice on-policy rollouts for the policy learning. The experiment results also look quite promising. I think the interesting part of the paper is that it combines several interesting ideas and manages to prove that they can significantly improve the performance of imitation learning in the challenging scenarios.

However, I still have certain concerns about the paper, which make me feel that the paper is not ready for publication yet. Please find the detailed comments below.


**Time Spent Reviewing:**

7

---

> ### Author Response · Authors · 2021-08-11
> **Response to  LYjg (part 1)**
>
> Thank you for your detailed review! Please find responses to your questions and comments below.
>
> **Q: Novelty and contribution**
>
> While some components of VMAIL exist in prior work, the paper describes and show several aspects of VMAIL that were not possible in prior works:
> - The ability to efficiently and reliably learn challenging vision-based tasks from demonstrations. Indeed, the experiments show much more stable training and higher asymptotic results over prior model-free algorithms.
> - The ability to transfer the model to learn new, but related tasks without any additional environment interaction. To our knowledge, no prior method has shown this capability.
>
> Based on your feedback we implemented an ablation to evaluate the individual contributions of variational representation learning and on-policy model-based training employed in VMAIL. For this purpose we train the DAC algorithm on top of the latent states from our model, which is only used for representation learning purposes. On the walker task, we find that this method matches the asymptotic performance  of VMAIL, **but requires 30% more data** (training curves are available on the project website, link is in the abstract). This demonstrates both the importance of representation learning for stability and the sample efficiency of model-based approaches.
>
> In addition, such model-free approaches do not allow us to train imitation policies for new tasks without generating additional data to train the discriminator, critic and policy, which is an innovation introduced in this work.
>
> **Q: Interpretation of Theorem 1 and connections to practical algorithm.**
>
> Thank you for the question and the opportunity to clarify. We believe there is a significant misunderstanding, and we hope to clarify below.
>
> Firstly, Theorem 1 shows that the divergence in the latent (belief) space is an upper bound on the divergence in the underlying state space. This is a very useful result, since it provides legitimacy to the idea of minimizing latent space divergence, since it will ultimately amount to optimizing a bound on the policy suboptimality (Section 3.1). Thus, we believe Theorem 1 is an important result that provides inspiration for our practical algorithm.
>
> Secondly, while the latent state space model captures the belief (latent space) representation as you mention, we can recover the belief visitation distribution by performing rollouts using the latent space dynamics model. The final objective we optimize is given in Eq. 6 (implemented using a value function as shown in Eq. 8), which exactly amounts to minimizing an f-divergence (Jensen-Shannon) between the policy and expert visitation distributions, as detailed in Sections 3.2 and 4.1.

---

> > ### Author Response · Authors · 2021-08-11
> > **Response to LYjg (part 2)**
> >
> > **Q: “However, not all IRL methods are adversarial imitation learning. In fact, most of them don’t train a GAN-based classifier and do RL afterwards. Instead, a lot of them recover the reward and do planning instead.”**
> >
> > Thank you for that out. We will change the language to make the separation clearer.
> >
> > **Q: “Why is this zero-shot?”**
> >
> > By “zero-shot” we mean that the agent can achieve good performance on a new imitation learning task with zero additional environment interaction that is usually required for adversarial IL. Note that demonstrations for the new task are unavoidable, since they are required to even define the new task. We will explicitly clarify in the next revision that we consider a transfer setting where a small number (non-zero) of expert demonstrations are provided and zero additional environment interactions are allowed.
> >
> > **Q: “In addition, I guess the transferability might be limited by the real difficulty of the source task / target task.”**
> >
> > Thank you for the suggestion, we will clarify the writing in the next revision.
> >
> > In this work, we are primarily concerned with transfer of behaviors where the source and target domains share the same (or very similar) dynamics. We expect good transfer from source to target domain when there exists good overlap between the replay buffer and the target domain demonstration. Note that overlap between source demonstration and target demonstration is not strictly needed. This is because the agent collects additional experience through online interactions in the source domain to build up the replay buffer, enabling the learning of a dynamics model with broader coverage than the source domain demonstrations.
> >
> > **References**
> >
> > [1] Ian J. Goodfellow, Jean Pouget-Abadie, M. Mirza, Bing Xu, David Warde-Farley, Sherjil Ozair, Aaron C. Courville, and Yoshua Bengio. Generative adversarial nets. In NIPS, 2014
> >
> > [2] Jonathan Ho and Stefano Ermon. Generative adversarial imitation learning. Conference on 398 Neural Information Processing Systems, 2016

---

> > ### Comment · Reviewer_LYjg · 2021-08-17
> > **thanks for the clarification**
> >
> > Thanks a lot for the detailed response and clarification to my questions. I agree that I had some misunderstandings about the correctness of Theorem 1 and several other claims. I've increased my rating to 6.

---

> ### Author Response · Authors · 2021-08-16
> **Thank you again for your review!**
>
> Thank you again for your review! Please let us know if there are still any remaining questions! We will be happy to elaborate further.

---

> ### Comment · Reviewer_K7pU · 2021-08-16
> **Questioning score and reason for rejection**
>
> Dear fellow reviewer LYjg,
>
> I could be wrong but it sounds like your main reason for your low rating is that this work is built upon existing components and I honestly don't think that this is grounds for rejection. I would argue that the majority of NeurIPS publications use some existing/published components.
>
> But I agree with the request for ablation studies that separate out the effect of the different components. That would be great to see here, but the underlying DREAMER paper has already addressed some of this.

---

> > ### Comment · Reviewer_LYjg · 2021-08-17
> > **editing the score after author response**
> >
> > Dear fellow reviewer K7pU,
> >
> > Thanks a lot for the notes. I agree the paper has presented interesting ideas and good results. Previously my main concern was about the correctness and certain claims of the work, but the authors' response has addressed it. I've updated my reviews accordingly.

---

### Official Review · Reviewer_K7pU · 2021-07-16

**Rating:** 8
**Confidence:** 4

**Summary:**

The authors present a novel method for imitation learning from a limited set of expert demonstrations. The method is heavily based on the DREAMER model-based RL framework and extends this with a discriminator to match a learner policy to an expert rollout.
I find the paper overall well-written and a worthwhile contribution but there are some technicalities holding the paper back from being a top submission. I am overall still in favor of getting this work accepted since I believe most things can be addressed easily and I'm happy to change my rating if the authors can address some of my concerns.

**Limitations And Societal Impact:**

I would like to commend the authors for being the only one in my 6 reviews that actually followed instructions and put the limitations into their own little subsection in the conclusion.
One additional limitation that I can see is resilience to visual distractors. Since your state is based on image to image reconstruction, all pixels, including meaningless but moving background and other actors will be reconstructed and therefore have to be captured in the latent state. Just a single distractor agent that's doing something task-independent could throw this method off, no?

**Main Review:**

**Originality:**

The work combines a GAIL-like discriminator between expert and agent trajectory with a much more sample-efficient world model-style approach, for which the authors used DREAMER. There are many other trajectory embedding approaches to imitation learning, most of which are cited or even used as a baseline but one noteworthy omission is VIRL [1], which has a similar network structure but lower sample efficiency. I think it would be good to at least acknowledge that work in the related works section.

**Quality:**

The method is explained in great detail but to a fault. In my personal opinion, the POMDP formulation is not necessary at all. While technically, visual control tasks in 3D should always be considered POMDPs, in most contemporary works, they aren't. And accordingly, I don't think you gained anything by including section 4.2 in the main body of the paper (rather than the appendix), and instead I would prefer more ablation studies.
- Specifically, I'd be curious to see the influence of the number of expert samples on the training performance.
- I also would like to see more than 3 seeds in the result plots since there is such an incredibly high variance. Please see this tutorial on picking a number of seeds that is statistically significant: https://arxiv.org/pdf/1806.08295.pdf
- I think there is some issue in the implementation of the baseline methods: Neither SQIL nor DAC are nearly as unstable in their original publications as they are depicted here, which leads me to believe that the author-added DrQ modification may have had negative impacts on stability. This needs to be addressed urgently.

**Clarity:**

The paper is clearly written and the authors get huge bonus points from me for including working code with instructions for running it.

**Significance:**

I think this work is a very meaningful extension of DREAMER and definitely a valuable new imitation learning method that does not need excessive environment interactions while learning. I didn't think about using model-based RL methods in this way, but I really like the idea.

**Nitpicks and questions:**
1. line 16 "better stability compared to prior work" - not at any statistical significance with only 3 seeds and not with the unexplained performance deterioration of the baselines
2. line 26 ...circumventing exploration... - it doesn't really circumvent exploration. If the expert demonstration only covers a tiny part of the state space, the imitation learning has no chance of generalizing outside of that.
3. Figure 1 right side is great and communicates all major points! The bold and dash lines on the left side are probably more confusing than anything. Maybe something like Fig 3a from the dreamer paper would be better.
4. About the zero-shot/few-shot: I think this is only valid when both tasks share mostly the same dynamics, so the forward rollout model is identical given a state and action. But if you're going from one task to the other, e.g. driving as fast as possible while avoiding obstacles on CarRacing vs. making a lot of sharp turns in CarRacing, then I don't think this holds up. This wasn't really explored in the transfer section and probably needs a sentence. In your examples, the environment dynamics of walker-walk are included in walker-run and the environment dynamics of the 4-pronged D'Claw clockwise are too similar to the other tasks.
5. line 237 "scale" - scale from what?
6. line 241 1000 GPU hours total or each or how are they distributed?
7. line 254 CarRacing-v0 is not terribly hard to solve and for example Ilya Kostrikov's PPO implementation (https://github.com/ikostrikov/pytorch-a2c-ppo-acktr-gail/tree/master/a2c_ppo_acktr) can get to a mean episodic reward of ~550 and 700 within 500k and 1M steps respectively.
8. Figure 3: "consistently outperforms" - Not sure this is correct, looking at the top performance, that's not true for Cheetah Run, and looking at the early performance, that's not true for Claw, Car Racing, and Walker.
9. line 295 how does that work if you have 2 different action spaces - 3-pronged D'Claw has a smaller action space than 4-pronged D'Claw, right?
10. line 334 "Finally, the experiments suggest that this algorithm is efficient enough to be applied to real robots," - I don't know how this sentence is justified.


**References:**

- [1] Berseth, Glen, and Christopher J. Pal. "Visual Imitation Learning with Recurrent Siamese Networks." arXiv preprint arXiv:1901.07186 (2019).), https://arxiv.org/abs/1901.07186

**Time Spent Reviewing:**

6

---

> ### Author Response · Authors · 2021-08-11
> **Response to K7pU (part 1)**
>
> Thank you very much for your detailed review! We appreciate your thorough and constructive feedback. We believe that the revisions based on your comments will improve the paper. Please find answers to your questions and updates on revisions below.
>
> **Q: Reg. VIRL[1].**
> Thank you for the reference; we will add VIRL to our related works section, and discuss it in the revised version.
>
> **Q: “Specifically, I'd be curious to see the influence of the number of expert samples on the training performance.”**
>
> The results presented in our main submission uses 10 expert trajectories for all tasks except Baoding balls (25 trajectories). **We ran additional ablations with 5 and 1 expert trajectories on the dm_control tasks**, and we have added these results to the project website (url can be obtained from the abstract). The results with 5 expert trajectories are similar to results presented in the main paper. While using a single expert trajectory is particularly challenging, VMAIL was still able to make meaningful progress, achieving nearly 80% of the expert performance.
>
> **Q: More random seeds for experiments**
>
> We ran additional random seeds (for a total of 6) for the DM Control tasks, which showed quantitatively the same performance at 500k steps (see below). We will add these results as well as additional random seeds for remaining experiments in the camera ready version.
>
> |         |      | VMAIL | SQIL   | DAC   |
> |---------|------|-------|--------|-------|
> | Cheetah | mean | 597.9 | 19.6   | 630.2 |
> |         | std  | 12.5  | 40.4   | 21.9  |
> | Walker  | mean | 917.0 | 305.9  | 356.7 |
> |         | std  | 9.6   | 160.26 | 279.7 |
>
> **Q: “Neither SQIL nor DAC are nearly as unstable in their original publications as they are depicted here”**
> After the original publication of these methods, some subsequent works have also reported instability of SQIL and DAC. In [1] the authors show instability in SQIL training on low-dimensional domains (including complete collapse in performance in some environments). In [2] (which became public after the submission deadline) the authors also claim that they were unable to train DAC from images in a stable manner. We are unrelated to the authors of these works.
>
> Based on your suggestion, we also ran an ablation that does not use DrQ augmentation, which resulted in substantially lower learning -- achieving only 20% of the performance of the variant using DrQ augmentations at the 200K timestep mark.

---

> > ### Author Response · Authors · 2021-08-11
> > **Response to K7pU (part 2)**
> >
> > **Q: “... Circumventing exploration...”**
> >
> > Our intention was to communicate that demonstrations directly specify the desired behavior for the agent, instead of the agent exploring the environment to find high reward states by chance (as commonly the case in RL). We agree with your comment that generalization and extrapolation from the demonstrations may not be directly possible. We will revise the language to make that more clear.
> >
> > **Q: Reg. Figure 1**
> >
> > Thank you for the feedback. We will remove dashed lines and update the figure for better clarity.
> >
> > **Q: Reg. Zero/Few-shot transfer**
> >
> > We agree that overlap between the replay buffer and expert policy in target domain is required for transfer. However, note that overlap between demonstrations in the source and target domain is not required. This is because the agent collects additional experience through online interactions in the source domain to build up the replay buffer, enabling the learning of a dynamics model with broader coverage than the source domain demonstrations.
> >
> > **Q: “line 237 "scale" - scale from what?”**
> > We have replaced “scale to” with “solve”.
> >
> > **Q: “line 241 1000 GPU hours total or each or how are they distributed?”**
> > This refers to the total compute time for the project. We will clarify this in the revised version.
> >
> > **Q: “Figure 3: consistently outperforms...”**
> > VMAIL matches or out-performs asymptotic returns of the prior methods. We will revise that language to make that clear.
> >
> > **Q: “line 295 two different action spaces?”**
> >
> > Both the 3 pronged valve and 4 pronged vale environments have the same robot configuration and action space with 3 fingers. Only the objects they interact with are different (3 vs 4 pronged valves). We will clarify this in the text. There are videos on the project website linked in the abstract for a visual description.
> >
> > **Q: “line 334 "Finally, the experiments suggest that this algorithm is efficient enough to be applied to real robots," - I don't know how this sentence is justified.”**
> >
> > We used this as our opinion or observation, and not a concrete claim. We will change the language to clarify. That said, we also note that 100k samples in the D’Claw platform corresponds to 3 hours of continuous operation [3], which is definitely within the realm of real-world possibilities. We also ran some preliminary hardware experiments, videos and description of which can be obtained from the project website (link in abstract).
> >
> > **References**
> >
> > [1] Of Moments and Matching: A Game-Theoretic Framework for Closing the Imitation Gap, Swamy et. al., https://arxiv.org/pdf/2103.03236.pdf
> >
> > [2] Q-attention: Enabling Efficient Learning for Vision-based Robotic Manipulation, James et. al., https://arxiv.org/abs/2105.14829
> >
> > [3] ROBEL: Robotics Benchmarks for Learning with Low-Cost Robots, Ahn et. al., https://arxiv.org/abs/1909.11639

---

> > > ### Comment · Reviewer_K7pU · 2021-08-16
> > > **Thanks for clarifications - updating score.**
> > >
> > > Dear Authors,
> > >
> > > Thanks for the clarifications and additional experiments, especially the ones with fewer experts. Good stuff!
> > >
> > > I'm bumping my rating up from 7 to 8.
> > >
> > > Cheers

---

### Official Review · Reviewer_uqtX · 2021-07-17

**Rating:** 7
**Confidence:** 4

**Summary:**

Motivated by several challenges that face the use of adversarial imitation learning (AIL) in practice, the authors propose V-MAIL—a model-based algorithm for visual AIL. V-MAIL addresses the challenges of sample inefficiency and difficult numerical optimization by learning and leveraging both a variational observation model and a variational forward dynamics model. The authors provide/prove interesting theoretical performance bounds with respect to the proposed formulation, and propose V-MAIL as a practical solution to that formulation.

**Limitations And Societal Impact:**

I found the authors’ treatment of these topics in the paper to be sufficient.

**Main Review:**

STRENGTHS:

(S1) The proposed approach is novel and proven effective empirically in standard benchmark domains.

(S2) The authors have provided a relevant and interesting theoretical analysis that motivates the proposed algorithm.

WEAKNESSES:

(W1) The paper is missing discussion of important related literature in imitation from observation (IfO). While the authors should draw connection to the area as a whole, they also need to discuss the similarities/differences with BCO [Torabi et al., 2018] in particular, since it’s extremely similar. For example, BCO _also_ contains a model-learning phase followed by an imitation phase. To be clear, the proposed algorithm is unique in its own right, the authors just need to address the close relationship here.

POST-DISCUSSION COMMENTS:

Thanks to the authors for agreeing to revise the paper to make the connection to IfO more explicit. I'm happy to continue recommending acceptance.

**Time Spent Reviewing:**

3

---

> ### Author Response · Authors · 2021-08-11
> **Response to uqtX**
>
> Thank you for taking valuable time to review our paper! We are glad that you found the approach novel, the algorithm empirically effective, and the theoretical results interesting and relevant. We address your main concern below.
>
> **Q: Similarities/differences with BCO [Torabi et al., 2018]**
>
> We agree that the problem of imitation from observation (IfO) is related, and we will discuss BCO and the area as a whole in the revised paper. As a short summary, BCO learns an inverse dynamics model to infer actions from sequences of expert observations and then carries out behaviour cloning. In contrast VMAIL learns a forward dynamics model to generate data for policy training using an adversarial approach. Modifying VMAIL to work with only observations would make for exciting future work, but outside the scope of current submission.

---

> > ### Comment · Reviewer_uqtX · 2021-08-31
> > **Thanks**
> >
> > Thanks to the authors for the response and for agreeing to address the related work in the revision of the paper. I’m happy to keep my score as it is, and have no further questions.

---

### Official Review · Reviewer_ESTA · 2021-07-23

**Rating:** 6
**Confidence:** 4

**Summary:**

The authors propose V-MAIL, a variational model-based imitation learning algorithm. The transition dynamics model is learned by optimizing a relevant variational lower bound under the assumption of partial observability, while discriminators for imitation learning are trained over the latent-action space. The empirical results show that V-MAIL outperforms baselines including SQIL and DAC (where the data-augmentation technique is applied for both baselines). The authors additionally check the generalization ability (transfer learning) for V-MAIL.

**Limitations And Societal Impact:**

No societal impact was concerend.

**Main Review:**

Although the proposed idea outperforms their baselines, I think more baselines and related works should have been considered, which makes me vote for rejection of this work.
- The submission is not the first work considering a latent representation in imitation learning, e.g., Gangwani et al., "Learning Belief Representations for Imitation Learning in POMDPs". In Gangwani et al., the inequalities similar to Theorem 1 of this submission have been considered.
- The submission is not the first work on image-based imitation learning, e.g., Pathak et al., "Zero-shot visual imitation", Torabi et al., "Imitation Learning from Video by Leveraging Proprioception", Liu et al., "Imitation from Observation: Learning to Imitate Behaviors from Raw Video via Context Translation", etc. I couldn't write all here, but I could easily find out vision-based imitation learning algorithms that refer to GAIL paper in Google Scholar. (FYI, my keywords were "visual based learning", "pixel", "image" under the papers citing GAIL.) While most of them didn't consider model-based imitation, I think comparing V-MAIL with model-free vision-based imitation learning is necessary. (If not possible, authors need to state why some of those submissions cannot be compared with V-MAIL.)
- The contribution seems marginal. The proposed method combines well-known approaches in vision-based RL to improve the performance of imitation learning.

**Time Spent Reviewing:**

5

---

> ### Author Response · Authors · 2021-08-11
> **Response to ESTA (part 1)**
>
> Thank you for reviewing our paper and sharing feedback. We address your concerns below. We are happy to continue discussions to address any additional questions or concerns.
>
> **Q1: “The submission is not the first work considering a latent representation in imitation learning, e.g., Gangwani et al.”**
>
> Thank you for the reference. We agree that this paper is relevant and we will cite and discuss it (particularly in the context of Theorem 1).
>
> As a summary, Gangwani et al. aims to learn an observational filter for **low-dimensional** observation spaces, which is subsequently used in conjunction with an off-policy model-free algorithm. On the other hand, our approach is model-based, deals with high-dimensional observation spaces, and also provides theoretical results.
>
> Based on your feedback, we conducted an ablation experiment using a method similar to Gangwani et al, where we train the DAC algorithm on top of the latent representation from our model. On the walker task, we find that this method matches the asymptotic performance  of VMAIL, **but requires 30% more data,** demonstrating the sample efficiency gains of our model-based VMAIL algorithm. We have added the results to the project website, the link can be found from the paper (https://sites.google.com/view/variational-mail). Moreover, model-free methods like Gangwani et al. do not allow us to train agents for new imitation tasks without generating additional data to train the discriminator, critic and policy (Section 5.2).
>
>
> **Q2: The submission is not the first work on image-based imitation learning. Connections to Pathak et al., Torabi et al., Liu et al.?**
>
> Thank you for mentioning these works! We will discuss them in the revised papers. Broadly speaking, while these papers are related work in the sense of “imitation learning from image observations”, the exact problem settings and scope differ from those of our paper, thereby making them not suitable as direct baselines. We expand on this below.
>
> - Pathak et al. is specific to goal-conditioned RL, and is not applicable to tasks and environments that are are not goal-oriented (such as those considered in the experiments of our paper)
> - Torabi et al. considers a setting similar to ours, but assumes access to robot proprioception, which makes the problem substantially simpler than learning from visual inputs alone. Further, they use an on-policy model-free approach (PPO), which is not competitive in terms of sample complexity. **We ran such an ablation in our setting (from image observations only) and it showed little progress in 200k environment steps.**
> - Liu et al. considers a different problem setting, where there is domain shift between the demonstrator and agent. The main contributions of Liu et al. pertain to this domain shift, which is not present in the environments we consider, thereby making it an orthogonal contribution.

---

> > ### Author Response · Authors · 2021-08-11
> > **Response to ESTA (part 2)**
> >
> > **Q3: The contribution seems marginal. The proposed method combines well-known approaches in vision-based RL to improve the performance of imitation learning.**
> >
> > While we agree that individual components of VMAIL have been explored in prior work, we believe their combination is novel, and it is this novel combination that leads to strong empirical results. For example, our results in the Walker and Cheetah task are comparable (or even better) in sample efficiency compared to DrQ and Dreamer, despite using the learned discriminator rewards as opposed to ground truth rewards. Providing a few demonstrations is often much easier than scripting detailed rewards, and thus we believe our results constitute a major research advancement.
> >
> > We reiterate that VMAIL shows several advances and capabilities where were not possible with prior (mostly model-free) approaches:
> > - The ability to efficiently and reliably learn challenging vision-based tasks from demonstrations. Indeed, the experiments show much more stable training and higher asymptotic results over prior model-free algorithms.
> > - The ability to transfer the model to learn new, but related tasks without any additional environment interaction. To our knowledge, no prior method has shown this capability.

---

> ### Author Response · Authors · 2021-08-16
> **Thank you for your review**
>
> Thank you again for your review! Please let us know if there are still any remaining questions! We will be happy to elaborate further.

---

> ### Comment · Reviewer_K7pU · 2021-08-21
> **Any concerns left unaddressed**
>
> Dear Reviewer ESTA,
>
> I found the work and the authors' response to my review very compelling.
> If there's anything left preventing you from recommending acceptance of this paper, please let me and the authors know. I'm happy to champion this work.
>
> Best,
> Reviewer K7pU

---

> ### Comment · Reviewer_ESTA · 2021-08-22
> **Increase my score and vote for acceptance.**
>
> I read through the authors' responses, and each of them carefully addressed my concerns on the comparison with baselines. I sincerely appreciate the authors' efforts for the extra results. I increased my score to 6.

---

### Author Response · Authors · 2021-08-16
**Thank you again for your reviews**

Thank you again for your reviews. In summary, we have:

1. Added ablations of the number of expert trajectories used.
2. Added an ablation of model-based training, which shows VMAIL is 30% more sample efficient than the model-free approach.
3.We have added the recommended references and adjusted some of the language of the paper.

Please let us know if there are still any remaining questions and whether these changes have fully addressed your concerns.

---

### Decision · Program_Chairs · 2021-09-27

**Decision:**

Accept (Poster)

**Comment:**

This paper presents V-MAIL, a model-based algorithm for visual Adversarial Imitation Learning. V-MAIL learns both a variational observation model and a variational forward dynamics model. The algorithm is motivated by theoretical performance bounds, proposing the algorithm as a practical solution to the formulation. All the reviewers evaluated positive about the paper, and voted for acceptance.